# Assessing service availability and readiness to manage Chronic Respiratory Diseases (CRDs) in Bangladesh

**Progga Paromita**[1]*, **Hasina Akhter Chowdhury**[2©], **Cinderella Akbar Mayaboti**[2©], **Shagoofa Rakhshanda**[2], **A. K. M. Fazlur Rahman**[2], **Md. Rizwanul Karim**[3], **Saidur Rahman Mashreky**[2¤]

**1** Kirtipasha Union Health and Family Welfare Centre, Jhalokathi Sadar Upazila, Barishal, Bangladesh, **2** Centre for Injury Prevention and Research, Bangladesh (CIPRB), Dhaka, Bangladesh, **3** Department of Non Communicable Disease Control, Directorate General of Health and Services, Dhaka, Bangladesh

© These authors contributed equally to this work.

¤ Current address: Department of Non-Communicable Diseases, Bangladesh University of Health Sciences (BUHS), Dhaka, Bangladesh

* paromitaprogga65@gmail.com

**Data Availability Statement:** All relevant data are within the manuscript and its Supporting Information files.

## Abstract

### Introduction

Chronic Respiratory Diseases (CRDs) are some of the most prevailing non-communicable diseases (NCDs) worldwide and cause three times higher morbidity and mortality in low- and middle-income countries (LMIC) than in developed nations. In Bangladesh, there is a dearth of data about the quality of CRD management in health facilities. This study aims to describe CRD service availability and readiness at all tiers of health facilities using the World Health Organization's (WHO) Service Availability and Readiness Assessment (SARA) tool.

### Methods

A cross-sectional study was conducted from December 2017 to June 2018 in a total of 262 health facilities in Bangladesh using the WHO SARA Standard Tool. Surveys were conducted with facility management personnel by trained data collectors using REDCap software. Descriptive statistics for the availability of CRD services were calculated. Composite scores for facility readiness (Readiness Index 'RI') were created which included four domains: staff and guideline, basic equipment, diagnostic capacity, and essential medicines. RI was calculated for each domain as the mean score of items expressed as a percentage. Indices were compared to a cutoff of 70% which means that a facility index above 70% is considered 'ready' to manage CRDs at that level. Data analysis was conducted using SPSS Vr 21.0.

### Results

It was found, tertiary hospitals were the only hospitals that surpassed the readiness index cutoff of 70%, indicating that they had adequate capacity and were ready to manage CRDs (RI 78.3%). The mean readiness scores for the other hospital tiers in descending order were

**Funding:** Prof AKM Fazlur Rahman as the Principal Investigator received the fund from Directorate General Health Services (DGHS) of Bangladesh, Ref No: DGHS/LD/NCDC/Procurement plan/ GOB (Service)/2017-18/ 539 Sp-02. DGHS provided technical support during the implementation of the study. URL: https://dghs.gov.bd/index.php/bd/.

**Competing interests:** The authors have declared that no competing interests exist.

District Hospitals (DH): 40.6%, Upazila Health Complexes (UHC): 33.3% and Private NGOs: 39.5%).

## Conclusion

Only tertiary care hospitals, constituting 3.1% of sampled health facilities, were found ready to manage CRD. Inadequate and unequal supplies of medicine as well as a lack of trained staff, guidelines on the diagnosis and treatment of CRDs, equipment, and diagnostic facilities contributed to low readiness index scores in all other tiers of health facilities.

## Introduction

Combating non-communicable diseases (NCDs) is a major global public health challenge due to its high morbidity and mortality rates. NCDs were responsible for more than two-thirds of global deaths in 2019,killing 41 million people in total [1, 2], 16 million of which were premature deaths [3]. NCDs are likely to cost the world economy $47 trillion over the next 20 years, representing 75% of global gross domestic product (GDP) [4].

Chronic respiratory diseases (CRDs) are one of the four most prevalent NCDs in the world [2]. Examples of CRDs include, asthma, chronic obstructive pulmonary disease (COPD), occupational lung diseases, sleep apnea syndrome, and pulmonary hypertension [5]. The burden of preventable CRDs has major adverse effects on the quality of life and disability of affected individuals where women, children, and the elderly are the most vulnerable [6]. Global Disability Adjusted Life Years (DALYs) of CRDs are high and rising. For example, the global mortality rate due to COPD increased by almost 11% from 1990 to 2015 and if this current rate continues, it will be the third leading cause of global death by 2030 [7].

Though Bangladesh has achieved noteworthy progress in its health sector throughout the past decades in comparison with its neighbors [8], it is still suffering from the increased burden of NCDs [9]. NCDs account for 61% of the total disease burden in Bangladesh [3]. A vast number of people in Bangladesh are suffering from CRDs, where the prevalence of asthma is 5.2% [10] and the prevalence of COPD is 12.5% [7]. The World Health Organization (WHO) recommends that NCD care should be integrated into Primary Health Care (PHC) to improve NCD service delivery [2]. In response to this recommendation, Bangladesh has set up NCD corners which provide prevention, screening, and treatment for common NCDs at the upazila/sub-district level in its three-tiered PHC system (sub-district/upazila, union, and village level) [11, 12]. However, there is still a huge gap in accessibility of NCD services in the primary and secondary levels of the health system due to an insufficient government budget for health, which accounts for only 3.4% of Bangladesh's GDP [12]. In addition to lack of government funding, NCD care in Bangladesh is limited due to poor supply chain management of essential medicines and other commodities [12]. Moreover, the health system does not offer most services in rural areas outside of Dhaka, meaning that trained staff and CRD services are inaccessible to patients there [8].

Additionally, there is paucity of data on NCD service readiness, availability, and utilization in primary care facilities due to lack of research in this field as well as weaknesses in the national management information system, DHIS2 [13]. Data on NCD service readiness, availability, and utilization in the facilities are crucial for governments and non-governmental organizations (NGOs) to implement appropriate interventions and to assess the quality and impact of the services provided [14]. The Service Availability and Readiness Assessment (SARA) is a comprehensive facility-based assessment and WHO SARA Standard Tool is used to collect

data for conducting the assessment [15]. The tool covers questions on essential services in health facilities, through which service 'availability' (whether facilities offer a variety of preventive and curative health services) and 'readiness' (whether facilities have the items required to deliver that service at the time of the site visit) can be evaluated rapidly [16]. Previously, studies have used the WHO SARA Standard Tool to assess other NCDs in Bangladesh, such as diabetes and cardiovascular disease (CVD) [8]. However, the WHO SARA Standard Tool has not been utilized to assess CRDs related services within all tiers of health facility levels of Bangladesh.

The Sustainable Development Goal Target 3.4 is to reduce by one third premature mortality from non-communicable diseases through prevention and treatment. Universal access to CRD services is an essential prerequisite for achieving this goal [15]. This study will help policy makers address gaps in current CRD service delivery which will further improve action plans for overall NCD prevention and control in Bangladesh.

## Methods

### Survey design

This cross-sectional sub-study was part of a larger study called Service Availability and Readiness Assessment (SARA) Survey for NCDs and Disability Service Delivery System in Bangladesh conducted from December 2017 to June 2018. For the purposes of this sub-study one variable for assessing availability and 27 variables for assessing readiness focusing on CRD related services were considered.

The WHO SARA Standard Tool collects information on the availability of medical equipment related to CRD, including their location and functional status and components of support systems (e.g., logistics, maintenance, and management).

### Sampling and recruitment

In Bangladesh, public facilities are established according to its administrative units (division, district, upazila, union) where tertiary, secondary, and primary facilities are situated in the divisions (medical college hospitals), districts (DH, MCWC), upazila (UHC), union (FWC, CC) respectively [17]. As the number of population living in administrative units is different, so the population served by each facility is not defined. It may be mentioned that UHCs provide 31% of government health service and each of it serves 100,000 to 400,000 people depending on its size [18].

To identify the total number of facilities at all three tiers in Bangladesh, the Bangladesh Health Facility Survey (BHFS) 2014, National Institute of Population Research and Training (NIPORT), Management Information System(MIS) Report of Directorate General of Health Services (DGHS), and records from director hospital administration DGHS were used [16]. From this, we identified19,184 total hospitals and health centers in the country [19].

From this total, the parent study, Service Availability and Readiness Assessment (SARA) Survey for NCDs and Disability Service Delivery System in Bangladesh, selected 590 facilities using a stratified random sampling procedure according to administrative units and level of facilities (**Fig 1**). CRD services are provided from tertiary level of health facilities upto upazila level health facilities in Bangladesh [8]. This includes UHCs, district hospitals, and medical college hospitals. Conversely, MCWCs and FWCs were excluded from this sub study since these facilities only offer family planning services, and not CRD services. Therefore, this sub study found a total of 273 facilities that provided CRD services. However, 11 more facilities were excluded from this sub study during the analysis of data. These facilities were excluded either because the questionnaires were incomplete or the hospitals were specialized in diseases other

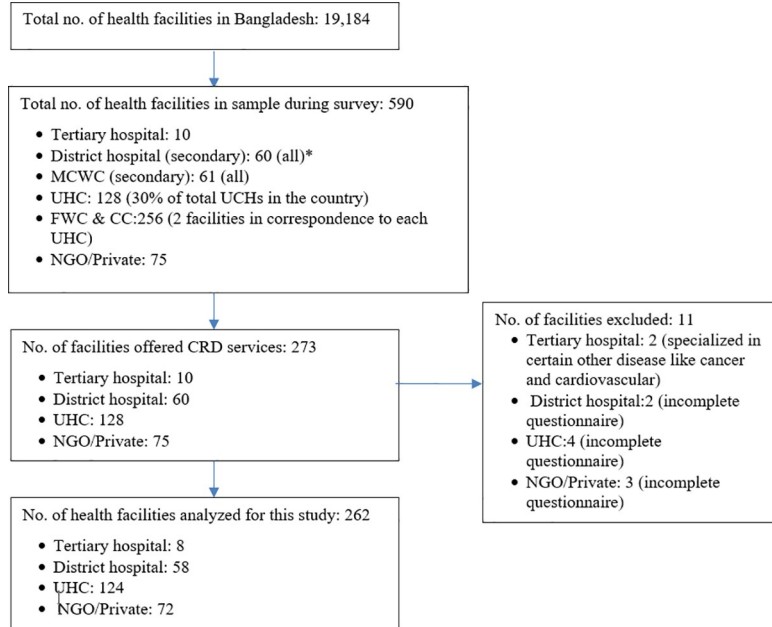

**Fig 1. Sampling and study inclusion flow chart.** N>B- UHC: Upazila Health Complex, MCWC: Maternal and Child Welfare Center, FWC: Family Welfare Center, CC: Community Clinic, NGO: Non Government Organization. *rest of the district hospitals had been converted to medical college hospital (tertiary facilities).

than CRD and were therefore irrelevant for this current study. The resulting sample size of 262 facilities is depicted in **Fig 1**.

The head of each facility or a management staff member who would have sufficient knowledge of facility capacity and operations were selected to participate in the survey (**Table 2**).

## Procedures

Forty-eight interviewers (MBBS graduates) were trained for two days to collect data using the survey tool and to enter data into REDCap using tablets. The health facilities sampled were contacted beforehand and an appointment was set with the focal person for interviewing.

## Data analysis

Descriptive analyses were performed using the domains (i.e. staff and guideline, equipment, diagnostic facilities and medicines) and corresponding readiness indicators/tracer items (**Table 1**).

Calculations for facility "availability" and "readiness" were based upon SARA guideline [20] and previous studies that utilized the SARA tool [5, 18]. Facility "availability" was assessed by asking if the health facility provided any CRD services (yes or no).

Mean availability and Service readiness were assessed in three stages: 1. Determining the mean availability of CRD service readiness indicators/tracer item at each facility level (number of facilities that have the tracer item available *100/ the total number of facilities); 2. Calculating the readiness index (RI) of facilities according to all 4 domains (the mean of all tracer item availability score in each domain);3. Calculating the facility level's overall readiness score (the average of the readiness index of all 4 domains). Indices were stratified by facility level and compared to an RI cutoff score of 70%. This cutoff was based upon a study conducted by Wilbroad Mutale *et al* in Zambia which utilized the SARA tool, where scores above 70% were

**Table 1. Tracer items in respective domains for CRD service.**

| Domains | Tracer Items |
|---|---|
| 1. Staff and guideline | Pulmonologist(full-time) |
| | Trained Nurse |
| | Physiotherapist |
| | Pulmonary technician |
| | CRD Guideline(available that day) |
| | Training on CRD management |
| 2. Equipments | Beds available for CRD patient |
| | Stethoscope |
| | Spacer for inhaler |
| | Oxygen delivery apparatus |
| 3. Diagnostic facility | Chest X-ray |
| | Sputum for gram stain, Culture and sensitivity(C/S), Acid-Fast Bacillus (AFB) |
| | Spirometry Test |
| | Peak flow meter |
| | Arterial blood gas (ABG) |
| | Sleep test |
| 4. Medicine | Beclomethasone inhaler |
| | Salmetrol + Fluticasone inhaler |
| | Dexamethasone injection |
| | Salbutamol inhaler |
| | Prednisolone |
| | Hydrocortisone |
| | Nebulizer solution |
| | Montelukast |
| | Tab Doxofylline |
| | Broad Spectrum Antibiotic |

considered as 'ready' to manage CRDs at that level [4]. All analyses were performed using SPSS Vr 21.0.

## Ethical considerations

Ethical clearance was obtained from the Ethical Review Board of the Center for Injury Prevention and Research, Bangladesh. Informed written consent was taken from each respondent.

## Results

### Health facility characteristics

A total of 262 health care facilities were assessed where 190 (72.5%) were public and 72 (27.5%) were from the private and NGO sector. The public facilities included primary [UHC -124(47.3%)], secondary [DH-58(22.1%)] and tertiary [8(3.1%)] level facilities. More than one fourth of facilities were situated in Dhaka division (**Table 2**)

### CRD service availability

The mean availability of CRD service is presented in all four domains under four types of facilities (primary, secondary, tertiary and private/NGO) (**Table 3**). Though tertiary hospitals exhibited higher availability of items than other facilities some of them had no pulmonologists

**Table 2. Overall characteristics of health facilities providing CRD services.**

| Facility Type | Interviewee type | Number of health facilities (n = 262) | % |
|---|---|---:|---:|
| Tertiary and Specialized hospital | Director or their representative | 8 | 3.1 |
| District hospital (DH) (Secondary level) | Civil Surgeon (CS)/Superintendent | 58 | 22.1 |
| Upazila Heath Complex (UHC) (Primary level) | Upazila Health and Family Planning Officer (UHFPO) or Resident Medical Officer (RMO) | 124 | 47.3 |
| NGO / Private hospital | Head of NGO/private hospital | 72 | 27.5 |
| **Ownership** | | | |
| Public | | 190 | 72.5 |
| Private | | 72 | 27.5 |
| **Division** | | | |
| Dhaka | | 70 | 26.7 |
| Barishal | | 22 | 8.4 |
| Chittagong | | 44 | 16.8 |
| Khulna | | 41 | 15.6 |
| Rajshahi | | 35 | 13.4 |
| Rangpur | | 34 | 13 |
| Sylhet | | 16 | 6.1 |

(25.0%) nor trained nurses (37.5%) for CRD treatment. In terms of basic equipment, all items were available in most of the facilities, except spacer for inhaler (DH-17.2%, UHC-8.1% Private-13.9%). In diagnostic facility domain, DH and UHC run neither ABG nor sleep test and 65% of the UHCs did not even have X-ray facilities. Except broad spectrum antibiotics, few essential medicines, especially inhalers, were available across facilities (**Table 3**).

## CRD service readiness

Readiness to manage CRDs varied widely among facilities surveyed (**Fig 2**). Tertiary hospitals had the highest mean readiness index score (78.4%) and were the only facility to cross the cutoff threshold of 70%, reflecting that they were the only facility level ready to provide CRD service. The rest of the public facilities scored far below the threshold [DH (40.7%), UHC (33.4%)]. The mean readiness index of NGO and private hospitals was also low at39.5% (**Fig 2**).

When disaggregated by domains, only tertiary facilities could reach the threshold index in any of the 4 domains. Rest of the facilities shown almost same score in their respective domains. Scores in the staff and guideline domain were low, with the lowest by NGO/Private (13.2%) (**Fig 3**).

## Discussion

The study reveals that, on average, the tertiary hospitals were ready to provide specific service for CRD patients, though there is still significant room for improvement. The rest of the health facilities throughout Bangladesh had inadequate CRD service availability and readiness.

Lack of essential commodities, including medicines and medical supplies, is a constant problem in public health facilities within Bangladesh [12], which was consistent with the limited supply of essential medicines for CRD management found in this study. In particular, supplies of spacers for inhaler and peak flowmeter supplies are extremely inadequate.

Existing literature illustrates that Bangladesh has a severe shortage of trained staff and inadequate diagnostic facilities [12], which was principally responsible for facilities' lower readiness scores in this study. For example, only 50% of the tertiary facilities have a pulmonary

**Table 3. Mean availability of CRD service readiness indicators in different levels of facilities (n = 262).**

| | Tertiary and Specialized hospital (n = 8) | District hospital (Secondary level) (n = 58) | Upazila Heath Complex (Primary level) (n = 124) | NGO / Private hospital (n = 72) |
|---|---|---|---|---|
| **Staff and guideline** | | | | |
| Pulmonologist(full time) | 6(75.0) | 0 | 0 | 4(5.6) |
| Trained Nurse | 5(62.5) | 3(5.2) | 5(4.0) | 3(4.2) |
| Physiotherapist | 5(62.5) | 2(3.4) | 0 | 6(8.3) |
| Pulmonary technician | 4(50.0) | 0 | 0 | 2(2.8) |
| CRD Guideline(available that day) | 7(87.5) | 38(65.5) | 70(56.5) | 32(44.4) |
| Training | 7(87.5) | 13(22.4) | 36(29.0) | 10(13.9) |
| **Equipment** | | | | |
| Beds available for CRD patient | 6(75.0) | 15(25.9) | 23(18.5) | 6(8.3) |
| Stethoscope | 8(100.0) | 58(100.0) | 124(100.0) | 72(100.0) |
| Spacer for inhaler | 5(62.5) | 10(17.2) | 10(8.1) | 10(13.9) |
| Oxygen delivery apparatus | 7(87.5) | 54(93.1) | 112(90.3) | 62(86.1) |
| **Diagnostic facility** | | | | |
| Chest X ray | 8(100.0) | 50(86.2) | 44(35.5) | 51(70.8) |
| Sputum for gram stain, Culture and sensitivity (C/S), Acid Fast Bacilli (AFB) | 8(100.0) | 46(79.3) | 98(79.0) | 44(61.1) |
| Spirometry Test | 8(100.0) | 11(19.0) | 12(9.7) | 16(22.2) |
| Peak flow meter | 6(75.0) | 6(10.3) | 9(7.3) | 9(12.5) |
| ABG | 6(75.0) | 0 | 0 | 5(6.9) |
| Sleep test | 5(62.5) | 0 | 0 | 3(4.2) |
| **Medicine** | | | | |
| Beclomethasone inhaler available | 4(50.0) | 11(19.0) | 5(4.0) | 35(48.6) |
| Salmetrol + Fluticasone inhaler | 4(50.0) | 11(19.0) | 5(4.0) | 39(54.2) |
| Dexamethasone injection | 8(100.0) | 38(65.5) | 75(60.5) | 46(63.9) |
| Salbutamol inhaler | 5(62.5) | 25(43.1) | 38(30.6) | 43(59.7) |
| Prednisolone | 7(87.5) | 29(50.0) | 31(25.0) | 38(52.8) |
| Hydrocortisone | 7(87.5) | 42(72.4) | 69(55.6) | 52(72.2) |
| Nebulizer solution | 8(100.0) | 44(75.9) | 94(75.8) | 56(77.8) |
| Montelukast | 8(100.0) | 39(67.2) | 74(59.7) | 44(61.1) |
| Tab Doxofylline | 7(87.5) | 18(31.0) | 29(23.4) | 35(48.6) |
| Broad Spectrum Antibiotic | 8(100.0) | 52(89.7) | 105(84.7) | 49(68.1) |

technician. The mean readiness index of staff and guideline for DH, UHC, and private hospitals are 16.1%, 15.3%, and 13.2% respectively, which are similar readiness index scores previously conducted using the SARA tool to assess CVD and diabetes at these facility levels [8]. As patient education reduces gaps in health care and disease management, so primary care physicians may be an effective target for interventions to improve CRD management through patient based simulation training in a resource poor country like Bangladesh [21].

While Leslie HH *et al.* reported, using the SARA Standard Tool, that private facilities in Bangladesh possessed higher RI for overall NCD services in contrast to public facilities [22], this was not found for CRDs in this study (39.5% RI). While according to the interviewee around 90% of DH& UHC (public facilities) and 80% NGO/private facilities provided CRD services to the patients, none of their readiness index scores indicated that they could sufficiently manage patients with CRD. Though Bangladesh's government established NCD corners in UHCs [11], these facilities' RI score (33.3%) was less than half of the threshold assessing readiness to treat CRDs (70%).

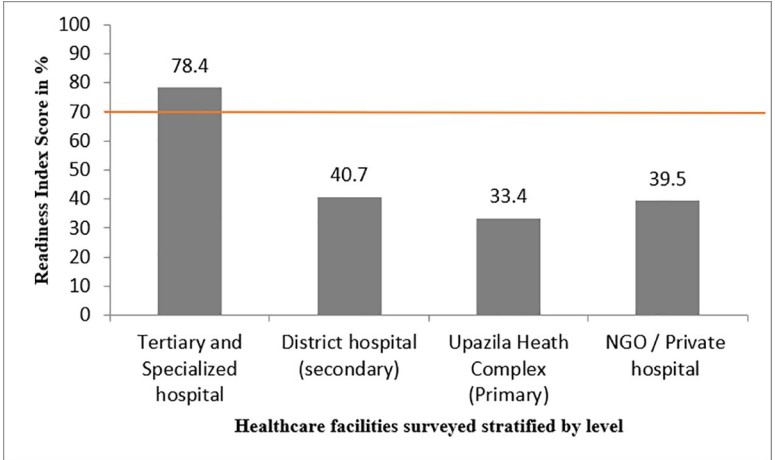

**Fig 2. Overall CRD service readiness index score for each facility level (the red line indicates the cut off value 70%, above which a facility is considered to be 'ready' to provide CRD services).**

The strength of this study is that we explored the current scenario of CRDs management in facilities throughout Bangladesh stratified by different levels. We explained it both in summarized and disaggregated in all four domains according to WHO SARA Standard Tool. However, there are some limitations to this study such as we could not include all the sample health facilities due to incomplete questionnaire and absence of proper interviewee. Data is based on reported information from the interviewed staff member and was not confirmed by observation. Moreover, we had to use same indicators for all level of facilities as there is no specific evidence mentioning what services should be provided at which level.

## Conclusion

In spite of significant development in the health sector, Bangladeshis still unable to provide effective management of CRDs. Almost all of the facilities observed in this study are below the

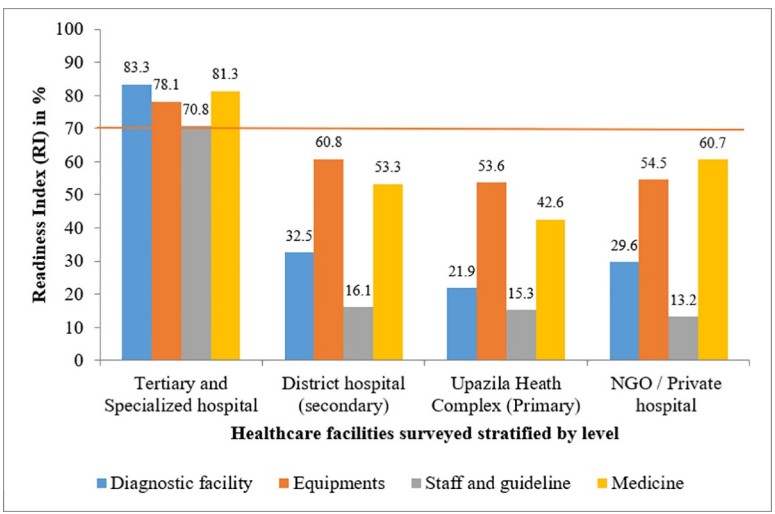

**Fig 3. Domain specific CRD service readiness index by facility level (the red line indicates the cut off value 70%, above which a facility is considered to be 'ready' to provide CRD services).**

readiness index threshold for CRD service delivery. Though tertiary hospitals, on average, are ready to provide CRD care, they constitute only 3.1% of sampled health facilities. This information can encourage policy makers and related stakeholders to improve current CRD service delivery approach; especially what minimum service should be provided at the primary and secondary healthcare levels, and advocate for equitable distribution of CRD services throughout Bangladesh.

## Supporting information

**S1 File.**
(SAV)

## Acknowledgments

We would like to acknowledge to the Non Communicable Disease Control (NCDC) Department of Directorate General of Health Services (DGHS) and CIPRB for making it possible through technical guidance and support. Last but not the least, our heartiest thanks to Sarah M. Anderson (MPH Candidate in Global Health, Rollins School of Public Health, Emory University) for proofreading and proving feedback on our drafts.

## Author Contributions

**Conceptualization:** A. K. M. Fazlur Rahman, Saidur Rahman Mashreky.

**Data curation:** Hasina Akhter Chowdhury.

**Formal analysis:** Progga Paromita, Saidur Rahman Mashreky.

**Funding acquisition:** A. K. M. Fazlur Rahman.

**Investigation:** Saidur Rahman Mashreky.

**Methodology:** A. K. M. Fazlur Rahman, Saidur Rahman Mashreky.

**Project administration:** A. K. M. Fazlur Rahman, Saidur Rahman Mashreky.

**Resources:** Saidur Rahman Mashreky.

**Supervision:** A. K. M. Fazlur Rahman, Saidur Rahman Mashreky.

**Validation:** Saidur Rahman Mashreky.

**Writing – original draft:** Progga Paromita.

**Writing – review & editing:** Hasina Akhter Chowdhury, Cinderella Akbar Mayaboti, Shagoofa Rakhshanda, Md. Rizwanul Karim, Saidur Rahman Mashreky.

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
