## [Decision Letter · Decision Letter 0]

19 Aug 2020

PONE-D-20-06779

Assessing Service Availability and Readiness to manage Chronic Respiratory Disease (CRD) in Bangladesh

PLOS ONE

Dear Dr. Paromita,

Thank you for submitting your manuscript to PLOS ONE. After careful consideration, we feel that it has merit but does not fully meet PLOS ONE’s publication criteria as it currently stands. Therefore, we invite you to submit a revised version of the manuscript that addresses the points raised during the review process.

The reviewers have provided a number of comments and suggestions regarding how the methods are described as well as the flow of the paper.  Please consider carefully these comments and provide point-by-point responses on how the comments were addressed. 

We look forward to receiving your revised manuscript.

Kind regards,

David Hotchkiss

Academic Editor

PLOS ONE

Journal Requirements:

3. Please include a copy of Table 1 which you refer to in your text on page 7.

Reviewers' comments:

Reviewer's Responses to Questions

**Comments to the Author**

1. Is the manuscript technically sound, and do the data support the conclusions?

Reviewer #1: Yes

Reviewer #2: Yes

Reviewer #3: No

2. Has the statistical analysis been performed appropriately and rigorously? 

Reviewer #1: Yes

Reviewer #2: Yes

Reviewer #3: No

3. Have the authors made all data underlying the findings in their manuscript fully available?

Reviewer #1: Yes

Reviewer #2: No

Reviewer #3: No

4. Is the manuscript presented in an intelligible fashion and written in standard English?

Reviewer #1: Yes

Reviewer #2: Yes

Reviewer #3: No

5. Review Comments to the Author

Reviewer #1: Respiratory Disease(CRD) management is an important step for medical care services. This manuscript is well written and pointed out an important gap to be filed to ensure proper CRD management. In the discussion authors suggested some guidelines, it would be good to see an additional point on Patient-Based Simulation Training. They can cite Cohen et al., 2014.

PMID: 24492338, DOI: 10.1097/SIH.0000000000000009

The overall quality and scientific findings meet the standard criteria for Plos One journal publications. I recommend to accept the manuscript for publishing.

Reviewer #2: 1. While the analysis is technically correct, the criteria for readiness are applied equally to all levels of facilities. However, lower level facilities would not be expected to provide the same level of services or have the same level of equipment and medicines as higher level facilities. The analysis should be revised to take into account the level of services expected or required at different levels of facilities.

3. There is no comment on the availability of data. The authors should state where or how the data could be accessed.

4. There are two versions of the manuscript provided. The second version is longer but superior to the first version, and my comments apply to the second version. Although the language in the second version is much better than in the first, a review for grammar and syntax is advised. eg line 60 prevailing should be prevalent. line 68 sector not sectors etc.

Further comments follow:

Introduction: It would be useful to describe here the organization of health facilities in Bangladesh, to assist in understanding the description in the methods section - which facilities are located at which administrative level, and population served; and to describe the two administrative wings of the MOH - health services and family planning. A brief description of the geographic organization would also assist as a basis for understanding the ‘divisions’ described in Table 2 eg a map and populations per division.

Methods:

More detailed description of the SARA methodology and sampling is provided compared to version 1 - which strengthens the paper.

Table 2 Would it be possible to include the total number of each level of facility as well as the number sampled, to provide a comparison with the sample size, and the extent to which it is representative of the distribution of facilities, at least for the public facilities.

Results

P10 line 162 ‘provided service’ – replace with ‘reported providing the service’ since the survey did not actually determine if the service was provided.

Figure 2 not required

Discussion

Some comment on the extent to which the sample is representative of facilities in Bangladesh is required.

It would appear that the survey assessed the same criteria for each level of service, with the same list of staffing, equipment, diagnostic facility and medicines. However, this does not take into account the particular services provided at each level of the health system. For example, it would not be expected that and FWC / community clinic could undertake spirometry, or would have a physiotherapist on staff.

It is recommended that the criteria for readiness be adjusted based on the services for CRD that are expected to be provided at each level, and then assess facilities at each level against the relevant criteria ?

A further limitation is that the data is based on reported information from the interviewed staff member, and was not confirmed by observation. This should be mentioned in limitations, as well as comment on the accuracy of the data collected by interview.

Conclusions

Rather than expect all services to be available at all levels, it would be important to develop a flow chart indicating what services should be provided at which level, and where referral to a higher level is needed for further care / investigation. Additional resources should then be provided to the level of facility that constitutes the greatest gap between required facilities and those reported.

Reviewer #3: Thank you for the opportunity to review this informative paper.This is an important issue, and I congratulate the authors for bringing further attention to it. The paper aimed to explore service availability and readiness at all tier of facilities in Bangladesh by using standard tool prepared by WHO. However, I have some slight amendments authors may wish to consider to improve the quality of the paper.

General comments

The topic is interesting and important for reducing the burden of CRD in LMIC. However, the manuscript needs extensive revision. The Introduction section missing a proper flow resulted to poor framing of research questions. In Methods, the sampling procedure and how the sample size has been derived is not clear. There is some redundancies of results in the Discussion section. Finally, the conclusion needs to be revised. If the authors revised these areas probably they would have a good article.

Below are my comments

--Abstract:

1. Generally, the flow of the abstract is not well structured, therefore, authors needs to revise it for clear understanding.

2. WHO SARA needs to be spelled out.

3. Methods; the second sentence “We collected data …….” Is not clear, authors need to revise it.

4. The last sentence not clear. What are the four domains? How availability and readiness index were calculated?

5. Results; only three out of two hundred facilities had adequate capacity……… it is not clear because in methods the total facilities were 537. What is RI mean?

6. The second sentence “Rests of ……….” Is not clear, what authors meant by far away from cut off value? How was it decided?

7. Conclusion; “………….only tertiary care hospitals………” this sentence is not true based on results. Authors should check it clearly. The conclusion does not tile with results.

--Introduction:

1. This Introduction section has rich of information, however, the flow is not proper i.e the authors should start and complete all information at Global level, then go to LMIC and then Bangladesh. Mixing and jumping of ides have observed in this section (for example; paragraph 1 included information from global, LMIC, and developed countries; paragraph 2 global and Bangladesh, and etc.).

2. 2nd and 3rd paragraph authors explained much about NCD rather than CRD which is the aim of this study.

3. Poor linking of information of NCD to CRD and availability of services/service provision made the authors failed to focus on the topic which lead to unclear explanation of their research question(s).

--Methods:

1. Generally, this section is not well structured.

2. Study design most information do not fall under this subsection. For example, training of interviewers, data collection and entry, and etc.

3. Sample size/ sampling technique; rather than providing the references the authors are required to explain in details the sample size calculation as well as sampling procedures.

4. Data analysis; surprisingly, most of the information written here not telling us how authors analyzed data. The details here are very vague and makes difficult to follow.

5. How the cut-off point of 70% reached? Is it based on the authors’ agreement without considering previous literature? If yes, why the authors did not choose to be 80% or 90% or 60%?

1. Ethical statement; This part is missing in the current version.

--Results:

1. CRD service availability; the text does not tile with the Figure 1 and 2.

2. CRD service readiness; what is the unit of readiness score? 2nd sentence “….. only 6 tertiary level hospitals out 537 were ready………” the use of 537 facilities in sentence is wrong or the comparison is wrong. The sentence is very confusing.

3. Missing of n in the all figures makes difficult to interpret data.

4. In all figures the test for significance is missing. Therefore, it is difficult to establish the statistical significance of the results.

--Discussion:

1. Generally, the discussion is not well written and the authors failed to discuss in detail the important findings from this study instead they summarized and presented the findings shown in the results section.

6. PLOS authors have the option to publish the peer review history of their article (what does this mean?). If published, this will include your full peer review and any attached files.

Reviewer #1: No

Reviewer #2: **Yes: **Krishna Philip Hort

Reviewer #3: No

---

## [Author Response · Author response to Decision Letter 0]

17 Oct 2020

REVIEWER #1

Comments: Respiratory Disease(CRD) management is an important step for medical care services. This manuscript is well written and pointed out an important gap to be filed to ensure proper CRD management. In the discussion authors suggested some guidelines, it would be good to see an additional point on Patient-Based Simulation Training. They can cite Cohen et al.,2014.PMID: 24492338, DOI: 10.1097/SIH.0000000000000009. The overall quality and scientific findings meet the standard criteria for Plos One journal publications. I recommend to accept the manuscript for publishing.

Response: Thank you so much for your feedback. We have gone through the literature you suggested and really it helped us to understand patient based simulation. We added it in our discussion as below-

‘Existing literature illustrates that Bangladesh has a severe shortage of trained staff and inadequate diagnostic facilities (12), which was principally responsible for facilities’ lower readiness scores in this study. For example, only 50% of the tertiary facilities have a pulmonary technician. The mean readiness index of staff and guideline for DH, UHC, and private hospitals are 16.1%, 15.3%, and 13.2 % respectively, which are similar readiness index scores previously conducted using the SARA tool to assess CVD and diabetes at these facility levels (8). As patient education reduces gaps in health care and disease management, so primary care physicians may be an effective target for interventions to improve CRD management through patient based simulation training in a resource poor country like Bangladesh (17).’

REVIEWER #2: 

Q1: While the analysis is technically correct, the criteria for readiness are applied equally to all levels of facilities. However, lower level facilities would not be expected to provide the same level of services or have the same level of equipment and medicines as higher level facilities. The analysis should be revised to take into account the level of services expected or required at different levels of facilities.

Response: Thank you so much for your feedback. We have addressed your concern in our Discussion (last paragraph) as below-

‘However, there are few limitations such as we could not include all the sample health facilities due to incomplete questionnaire and absence of proper interviewee. Data is based on reported information from the interviewed staff member and was not confirmed by observation. Moreover, we used same indicators for all level of facilities as there is no specific evidence mentioning what services should be provided at which level. Also, changes over time could not be observed as it is a cross-sectional study. The available data only reflects on the snapshot of availability and readiness.’

Q3: There is no comment on the availability of data. The authors should state where or how the data could be accessed.

Response: Thank you so much for your feedback.

The dataset has been uploaded.

Q4:There are two versions of the manuscript provided. The second version is longer but superior to the first version, and my comments apply to the second version. Although the language in the second version is much better than in the first, a review for grammar and syntax is advised. eg line 60 prevailing should be prevalent. line 68 sector not sectors etc.

Response: Thank you so much for your feedback.

We have corrected in line 60, 68 and also in 155 as below-

‘Chronic respiratory diseases (CRDs) are one of the four most prevalent NCDs in the world (2).’

‘Though Bangladesh has achieved noteworthy progress in its health sector throughout the past decades in comparison with its neighbors’

‘A total of 537 health care facilities were assessed where 465 (86.6%) were public and 72 (13.4%) were from the private and NGO sector’

Further comments follow:

Comments on Introduction: It would be useful to describe here the organization of health facilities in Bangladesh, to assist in understanding the description in the methods section - which facilities are located at which administrative level, and population served; and to describe the two administrative wings of the MOH - health services and family planning. A brief description of the geographic organization would also assist as a basis for understanding the ‘divisions’ described in Table 2 eg a map and populations per division.

Response: Thank you so much for your feedback. We have addressed your concern in our Introduction (3rd paragraph) as below-

‘Though Bangladesh has achieved noteworthy progress in its health sector throughout the past decades in comparison with its neighbors(8),it is still suffering from the increased burden of NCDs(9). NCDs account for 61% of the total disease burden in Bangladesh(3). A vast number of people in Bangladesh are suffering from CRDs, where the prevalence of asthma is 5.2% (10) and the prevalence of COPD is 12.5% (7).The World Health Organization (WHO)recommends that NCD care should be integrated into Primary Health Care (PHC) to improve NCD service delivery (2). In response to this recommendation, Bangladesh has set up NCD corners which provide prevention, screening, and treatment for common NCDs at the Upazila sub-district level in its three-tiered PHC system (sub-district, union, and village level)(11)(12).However, there is still a huge gap in accessibility of NCD services in the primary and secondary levels of the health system due to an insufficient government budget for health, which accounts for only 3.4 % of Bangladesh’s GDP (12). In addition to lack of government funding, NCD care in Bangladesh is limited due to poor supply chain management of essential medicines and other commodities (12). Moreover, the health system does not offer most services in rural areas outside of Dhaka, meaning that trained staff and CRD services are inaccessible to patients there(8)’

Comments on Methods Section:More detailed description of the SARA methodology and sampling is provided compared to version 1 - which strengthens the paper.Table 2 Would it be possible to include the total number of each level of facility as well as the number sampled, to provide a comparison with the sample size, and the extent to which it is representative of the distribution of facilities, at least for the public facilities.

Response: Thank you so much for your feedback. It has been edited and explained in detail (line 114 to 128). Added figure below (Fig 1)-

Fig 1: Sampling and study inclusion flow chart

Comments on Results: P10 line 162 ‘provided service’ – replace with ‘reported providing the service’ since the survey did not actually determine if the service was provided.Figure 2 not required.

Response: Thank you so much for your feedback. P10 line 162 ‘provided service’ –has been replaced as below-

‘Among all the facilities, 47% reported providing the service to CRD patients (Fig 2).’

Comments on Discussion:

1. Some comment on the extent to which the sample is representative of facilities in Bangladesh is required.It would appear that the survey assessed the same criteria for each level of service, with the same list of staffing, equipment, diagnostic facility and medicines. However, this does not take into account the particular services provided at each level of the health system. For example, it would not be expected that and FWC / community clinic could undertake spirometry, or would have a physiotherapist on staff.

Response:

Thank you so much for your feedback. We have addressed your concern in our Discussion (last paragraph) as below-

‘However, there are few limitations such as we could not include all the sample health facilities due to incomplete questionnaire and absence of proper interviewee. Data is based on reported information from the interviewed staff member and was not confirmed by observation. Moreover, we used same indicators for all level of facilities as there is no specific evidence mentioning what services should be provided at which level. Also, changes over time could not be observed as it is a cross-sectional study. The available data only reflects on the snapshot of availability and readiness.’

2. A further limitation is that the data is based on reported information from the interviewed staff member, and was not confirmed by observation. This should be mentioned in limitations, as well as comment on the accuracy of the data collected by interview.

Response: Thank you so much for your feedback. We have mentioned it in limitation (line 220) as below-

‘Data is based on reported information from the interviewed staff member and was not confirmed by observation.’

Comments on Conclusions: Rather than expect all services to be available at all levels, it would be important to develop a flow chart indicating what services should be provided at which level, and where referral to a higher level is needed for further care / investigation. Additional resources should then be provided to the level of facility that constitutes the greatest gap between required facilities and those reported.

Response: Thank you so much for your feedback. We have included this in our limitation and conclusion as below-

‘However, there are few limitations such as we could not include all the sample health facilities due to incomplete questionnaire and absence of proper interviewee. Data is based on reported information from the interviewed staff member and was not confirmed by observation.Moreover, we had to use same indicators for all level of facilities as there is no specific evidence mentioning what services should be provided at which level. Also, changes over time could not be observed as it is a cross-sectional study. The available data only reflects on the snapshot of availability and readiness.

Conclusion

In spite of significant development in the health sector, Bangladeshis still unable to provide effective management of CRDs. Almost all of the facilities observed in this study are below the readiness index threshold for CRD service delivery. Though tertiary hospitals, on average, are ready to provide CRD care, they constitute only 1.5% of total health facilities. This information can encourage policy makers and related stakeholders to improve current CRD service delivery approaches, especially what minimum service should be provided at the primary and secondary healthcare levels, and advocate for equitable distribution of CRD services throughout Bangladesh.’

REVIEWER #3: 

Thank you for the opportunity to review this informative paper.This is an important issue, and I congratulate the authors for bringing further attention to it. The paper aimed to explore service availability and readiness at all tier of facilities in Bangladesh by using standard tool prepared by WHO. However, I have some slight amendments authors may wish to consider to improve the quality of the paper.

Response: Thank you so much for your feedback

General comments: The topic is interesting and important for reducing the burden of CRD in LMIC. However, the manuscript needs extensive revision. The Introduction section missing a proper flow resulted to poor framing of research questions. In Methods, the sampling procedure and how the sample size has been derived is not clear. There is some redundancies of results in the Discussion section. Finally, the conclusion needs to be revised. If the authors revised these areas probably they would have a good article.

Response: Thank you so much for your feedback. We have revised the whole paper including 

Introduction: edited (line 78 to 98) as below-

‘However, there is still a huge gap in accessibility of NCD services in the primary and secondary levels of the health system due to an insufficient government budget for health, which accounts for only 3.4 % of Bangladesh’s GDP (12). In addition to lack of government funding, NCD care in Bangladesh is limited due to poor supply chain management of essential medicines and other commodities (12). Moreover, the health system does not offer most services in rural areas outside of Dhaka, meaning that trained staff and CRD services are inaccessible to patients there(8)

Additionally, there is paucity of data on NCD service readiness, availability, and utilization in primary care facilities due to lack of research in this field as well as weaknesses in the national management information system, DHIS2 (13).Data on NCD service readiness, availability, and utilization in the facilities are crucial for governments and non-governmental organizations (NGOs) to implement appropriate interventions and to assess the quality and impact of the services provided(14). The Service Availability and Readiness Assessment (SARA) is a comprehensive facility-based assessment and WHO SARA Standard Tool is used to collect data for conducting the assessment(15). The tool covers questions on essential services in health facilities, through which service ‘availability’ (whether facilities offer a variety of preventive and curative health services) and ‘readiness’ (whether facilities have the items required to deliver that service at the time of the site visit) can be evaluated rapidly(16). Previously, studies have used the WHO SARA Standard Tool to assess other NCDs in Bangladesh, such as diabetes and cardiovascular disease (CVD)(8). However, WHO SARA Standard Tool has not been utilized to assess CRDs related services within all health facility levels of Bangladesh.’

Methods: edited (line 120 to 136) (Fig 1) as below-

‘From this total, the parent study, Service Availability and Readiness Assessment (SARA) Survey for NCDs and Disability Service Delivery System in Bangladesh, selected10 tertiary and specialized hospitals for NCDs, all 60 district hospitals and 61 Mother and Child Welfare Centre, 128 UpazilaHealth Complexes (UHC) (two per district),256 Family Welfare Centres(FWC)& Community Clinics,and 75 NGO and private hospitals. 

This sub-study then excluded 53 health facilities for analysis: specialized hospitals that did not focus on CRD care, such as those devoted solely to cancer; facilities that did not fully complete the questionnaire, and hospitals where no staff member was able to complete the survey.This resulted in a final sample size of 537 facilities depicted in Fig1. 

Discussion and Conclusion: edited as below-

‘Discussion

The study reveals that, on average, the tertiary hospitals were ready to provide specific service for CRD patients, though there is still significant room for improvement. The rest of the health facilities throughout Bangladesh had inadequate CRD service availability and readiness.

Lack of essential commodities, including medicines and medical supplies, is a constant problem in public health facilities within Bangladesh (12), which was consistent with the limited supply of essential medicines for CRD management found in this study. In particular, supplies of spacers for inhaler and pick flow meter supplies are extremely inadequate.

Existing literature illustrates that Bangladesh has a severe shortage of trained staff and inadequate diagnostic facilities (12), which was principally responsible for facilities’ lower readiness scores in this study. For example, only 50% of the tertiary facilities have a pulmonary technician. The mean readiness index of staff and guideline for DH, UHC, and private hospitals are 16.1%, 15.3%, and 13.2 % respectively, which are similar readiness index scores previously conducted using the SARA tool to assess CVD and diabetes at these facility levels (8). 

While Leslie HH et al.reported, using the SARA Standard Tool, that private facilities in Bangladesh possessed higher RI for overall NCD services in contrast to public facilities (17),this was not true for CRDs in this study (39.5% RI). While according to the interviewee around 90% of DH& UHC (public facilities) and 80% NGO/private facilities provided CRD services to the patients, none of their readiness index scores indicated that they could sufficiently manage patients with CRD. Though Bangladesh’s government established NCD corners in UHCs (11), these facilities’ RI score (33.3%) was less than half of the threshold assessing readiness to treat CRDs (70%).

Almost every indicator was low in the MCWCs and FWC/CCs, which was not wholly surprising. MCWCs and FWCs fall under the Family Planning Wing of the Ministry of Health and Family Welfare, which focuses primarily on maternal and child healthcare, limiting their capacity for CRD care (18). Community Clinics are at the lowest level of healthcare delivery system and do not have adequate supplies nor manpower to provide CRD care to patients, often referring patients to facilities at higher tiers of the PHC system instead (18)(19).

The strength of this study is that we explored the current scenario of CRDs management in facilities throughout Bangladesh stratified by different levels. We explained it both in summarized and disaggregated in all four domains according to WHO SARA Standard Tool. However, there are few limitations such as we could not include all the sample health facilities due to incomplete questionnaire and absence of proper interviewee. Data is based on reported information from the interviewed staff member and was not confirmed by observation.Moreover, we had to use same indicators for all level of facilities as there is no specific evidence mentioning what services should be provided at which level. Also, changes over time could not be observed as it is a cross-sectional study. The available data only reflects on the snapshot of availability and readiness.

Conclusion

In spite of significant development in the health sector, Bangladeshis still unable to provide effective management of CRDs. Almost all of the facilities observed in this study are below the readiness index threshold for CRD service delivery. Though tertiary hospitals, on average, are ready to provide CRD care, they constitute only 1.5% of total health facilities. This information can encourage policy makers and related stakeholders to improve current CRD service delivery approaches, especially what minimum service should be provided at the primary and secondary healthcare levels, and advocate for equitable distribution of CRD services throughout Bangladesh.’

Abstract:

1. Generally, the flow of the abstract is not well structured, therefore, authors needs to revise it for clear understanding.

Response: Thank you so much for your feedback. Whole section is revised as below-

‘Abstract

Introduction: Chronic Respiratory Diseases (CRDs)are some of the most prevailing non-communicable diseases (NCDs) worldwide and cause three times higher morbidity and mortality inlow- and middle-income countries (LMIC) than in developed nations.In Bangladesh, there is a dearth of data about the quality of CRD management in health facilities. This study aims to describe CRD service availability and readiness at all tiers of health facilities using theWorld Health Organization’s (WHO) Service Availability and Readiness Assessment (SARA)tool.

Methods:A cross-sectional study was conducted from December 2017 to June 2018 in a total of 537 health facilities in Bangladeshusing the WHO SARAStandard Tool.Surveys were conducted with facility management personnel by trained data collectors using REDCapsoftware. Descriptive statisticsforthe availabilityof CRD services were calculated. Composite scoresfor facility readiness (Readiness Index ‘RI’) were created which included four domains: staff and guideline, basic equipment, diagnostic capacity, and essential medicines. RI was calculated for each domain as the mean score of itemsexpressed as a percentage. Indices were compared to a cutoff of 70% which means that a facility indexabove 70% is considered ‘ready’ to manage CRDs at that level. Data analysis was conducted using SPSS Vr 21.0.

Results: It was found,47% of all facilities reported providing CRD services. On average, tertiary hospitals were the only hospitals that surpassed the readiness index cutoff of 70%, indicating that they had adequate capacity and were ready to manage CRDs (RI 78.3%). The mean readiness scores for the other hospital tiers in descending order were District Hospitals (DH): 40.6%, Upazila Health Complexes (UHC): 33.3%, Private NGOs: 39.5%, Maternal and Child Welfare Centres (MCWC): 16.2% and Family Welfare Centres/Community Clinics (FWC/CC): 8.2%.

Conclusion: Only tertiary care hospitals, constituting 1.5% of all health facilities, were found ready to manage CRD. Inadequate and unequal supplies of medicine as well as a lack of trained staff, guidelines on the diagnosis and treatment of CRDs, equipment, and diagnostic facilities contributed to low readiness index scores in all other tiers of health facilities.’

2. WHO SARA needs to be spelled out.

Response: Thank you so much for your feedback. It has been spelled out (line 34) as below-

‘This study aims to describe CRD service availability and readiness at all tiers of health facilities using theWorld Health Organization’s (WHO) Service Availability and Readiness Assessment (SARA)tool.’

3. Methods; the second sentence “We collected data …….” Is not clear, authors need to revise it.

Response: Thank you so much for your feedback. It has been edited (line 35 to 38) as below-

‘A cross-sectional study was conducted from December 2017 to June 2018 in a total of 537 health facilities in Bangladeshusing the WHO SARAStandard Tool.Surveys were conducted with facility management personnel by trained data collectors using REDCapsoftware.’

4. The last sentence not clear. What are the four domains? How availability and readiness index were calculated?

Response: Thank you so much for your feedback. It has been edited (line 39 to 43) as below-

‘Descriptive statisticsforthe availabilityof CRD services were calculated. Composite scoresfor facility readiness (Readiness Index ‘RI’) were created which included four domains: staff and guideline, basic equipment, diagnostic capacity, and essential medicines. RI was calculated for each domain as the mean score of itemsexpressed as a percentage. Indices were compared to a cutoff of 70% which means that a facility indexabove 70% is considered ‘ready’ to manage CRDs at that level. Data analysis was conducted using SPSS Vr 21.0.’

5. Results; only three out of two hundred facilities had adequate capacity……… it is not clear because in methods the total facilities were 537. What is RI mean?

Response: Thank you so much for your feedback. It has been edited (line 44 to 49) as below-

‘Composite scoresfor facility readiness (Readiness Index ‘RI’) were created which included four domains: staff and guideline, basic equipment, diagnostic capacity, and essential medicines. RI was calculated for each domain as the mean score of itemsexpressed as a percentage.’

6. The second sentence “Rests of ……….” Is not clear, what authors meant by far away from cut off value? How was it decided?

Response: Thank you so much for your feedback. It has been edited (line 44 to 49), (line 144 to 152) as below-

‘It was found,47% of all facilities reported providing CRD services. On average, tertiary hospitals were the only hospitals that surpassed the readiness index cutoff of 70%, indicating that they had adequate capacity and were ready to manage CRDs (RI 78.3%). The mean readiness scores for the other hospital tiers in descending order were District Hospitals (DH): 40.6%, Upazila Health Complexes (UHC): 33.3%, Private NGOs: 39.5%, Maternal and Child Welfare Centres (MCWC): 16.2% and Family Welfare Centres/Community Clinics (FWC/CC): 8.2%.’

‘Service readiness was assessed in four stages: 1. Determining the availability of CRD service readiness indicators at each facility level; 2. Calculating the tracer item indexscores (number of tracer item present *100/number of tracer item should be present); 3. Calculating the readiness index (RI) of facilities according to all 4 domains (the mean of all tracer item index scores in each domain); 4. Calculating the facility level’s overall readiness score (the average of the readiness index of all 4 domains). Indices were stratified by facility level and compared to an RI cutoff score of 70%. This cutoff was based upon a study conducted by Wilbroad Mutale et al in Zambia which utilized the SARA tool, where scores above 70% were considered as ‘ready’ to

 manage CRDs at that level (5). All analyses were performed using SPSS Vr 21.0.’

7. Conclusion; “………….only tertiary care hospitals………” this sentence is not true based on results. Authors should check it clearly. The conclusion does not tile with results.

Response: Thank you so much for your feedback. We have revised and edited as below-

Result: ‘On average, tertiary hospitals were the only hospitals that surpassed the readiness index cutoff of 70%’ (line 44, 45)

Conclusion: ‘Only tertiary care hospitals, constituting 1.5% of all health facilities, were found ready to manage CRD’ (line 50, 51)

--Introduction:

1. This Introduction section has rich of information, however, the flow is not proper i.e the authors should start and complete all information at Global level, then go to LMIC and then Bangladesh. Mixing and jumping of ides have observed in this section (for example; paragraph 1 included information from global, LMIC, and developed countries; paragraph 2 global and Bangladesh, and etc.).

Response: Thank you so much for your feedback. It has been revised and edited as below-

Paragraph 1&2 – Global information. First on NCD in paragraph 1 and then on CRD in paragraph 2

Paragraph 3&4- NCD and CRD situation in Bangladesh

Paragraph 5- Gaps

Paragraph 6: Objectives

‘Introduction

Combating non-communicable diseases (NCDs) is a major global public health challenge due to its high morbidity and mortality rates. NCDs were responsible for more than two-thirds of global deaths in 2019,killing 41 million people in total (1)(2), 16 million of which were premature deaths (3). NCDs are likely to cost the world economy $47 trillion over the next 20 years, representing 75% of global gross domestic product (GDP)(4).

Chronic respiratory diseases (CRDs) are one of the four most prevalent NCDs in the world (2). Examples of CRDs include, asthma, chronic obstructive pulmonary disease (COPD), occupational lung diseases, sleep apnea syndrome, and pulmonary hypertension(5). The burden of preventable CRDs has major adverse effects on the quality of life and disability of affected individuals where women, children, and the elderly are the most vulnerable(6).Global Disability Adjusted Life Years (DALYs) of CRDs are high and rising. For example, the global mortality rate due to COPD increased by almost 11% from 1990 to 2015 and if this current rate continues, it will be the third leading cause of global death by 2030 (7). 

Though Bangladesh has achieved noteworthy progress in its health sector throughout the past decades in comparison with its neighbors(8),it is still suffering from the increased burden of NCDs(9). NCDs account for 61% of the total disease burden in Bangladesh(3). A vast number of people in Bangladesh are suffering from CRDs, where the prevalence of asthma is 5.2% (10) and the prevalence of COPD is 12.5% (7).The World Health Organization (WHO)recommends that NCD care should be integrated into Primary Health Care (PHC) to improve NCD service delivery (2). In response to this recommendation, Bangladesh has set up NCD corners which provide prevention, screening, and treatment for common NCDs at the Upazila sub-district level in its three-tiered PHC system (sub-district, union, and village level)(11)(12).However, there is still a huge gap in accessibility of NCD services in the primary and secondary levels of the health system due to an insufficient government budget for health, which accounts for only 3.4 % of Bangladesh’s GDP (12). In addition to lack of government funding, NCD care in Bangladesh is limited due to poor supply chain management of essential medicines and other commodities (12). Moreover, the health system does not offer most services in rural areas outside of Dhaka, meaning that trained staff and CRD services are inaccessible to patients there(8)

Additionally, there is paucity of data on NCD service readiness, availability, and utilization in primary care facilities due to lack of research in this field as well as weaknesses in the national management information system, DHIS2 (13).Data on NCD service readiness, availability, and utilization in the facilities are crucial for governments and non-governmental organizations (NGOs) to implement appropriate interventions and to assess the quality and impact of the services provided(14). The Service Availability and Readiness Assessment (SARA) is a comprehensive facility-based assessment and WHO SARA Standard Tool is used to collect data for conducting the assessment(15). The tool covers questions on essential services in health facilities, through which service ‘availability’ (whether facilities offer a variety of preventive and curative health services) and ‘readiness’ (whether facilities have the items required to deliver that service at the time of the site visit) can be evaluated rapidly(16). Previously, studies have used the WHO SARA Standard Tool to assess other NCDs in Bangladesh, such as diabetes and cardiovascular disease (CVD)(8). However, WHO SARA Standard Tool has not been utilized to assess CRDs related services within all health facility levels of Bangladesh. 

The Sustainable Development Goal Target 3.4 is to reduce one third premature mortality from non-communicable diseases through prevention and treatment. Universal access to CRD services is an essential prerequisite for achieving this goal.(15).This study will help policy makers address gaps in current CRD service delivery which will further improve action plans for overall NCD prevention and control in Bangladesh.’

2. 2nd and 3rd paragraph authors explained much about NCD rather than CRD which is the aim of this study.

Response: Thank you so much for your feedback. 2nd paragraph was revised fully and now explained the CRD information globally.

In 3rd paragraph we explained our Bangladesh scenario. But we did not have much information specifically on CRD service in Bangladesh. Rather it is actually merged with NCD sevices. That’s why we had to use information on NCD mostly on Bangladesh perspective. 

We have mentioned in line 94 to 98 ‘Previously, studies have used the WHO SARA Standard Tool to assess other NCDs in Bangladesh, such as diabetes and cardiovascular disease (CVD)(8). However, WHO SARA Standard Tool has not been utilized to assess CRDs related services within all health facility levels of Bangladesh’

3. Poor linking of information of NCD to CRD and availability of services/service provision made the authors failed to focus on the topic which lead to unclear explanation of their research question(s).

Response: Thank you so much for your feedback. It has been revised and edited as below-

Paragraph 1 –Global information on NCD 

Paragraph 2- How CRD is linked to NCD and its Global perspective explained

Paragraph 3&4- NCD and CRD situation in Bangladesh

Paragraph 5- Gaps

Paragraph 6: Objectives. 

‘Introduction

Combating non-communicable diseases (NCDs) is a major global public health challenge due to its high morbidity and mortality rates. NCDs were responsible for more than two-thirds of global deaths in 2019,killing 41 million people in total (1)(2), 16 million of which were premature deaths (3). NCDs are likely to cost the world economy $47 trillion over the next 20 years, representing 75% of global gross domestic product (GDP)(4).

Chronic respiratory diseases (CRDs) are one of the four most prevalent NCDs in the world (2). Examples of CRDs include, asthma, chronic obstructive pulmonary disease (COPD), occupational lung diseases, sleep apnea syndrome, and pulmonary hypertension(5). The burden of preventable CRDs has major adverse effects on the quality of life and disability of affected individuals where women, children, and the elderly are the most vulnerable(6).Global Disability Adjusted Life Years (DALYs) of CRDs are high and rising. For example, the global mortality rate due to COPD increased by almost 11% from 1990 to 2015 and if this current rate continues, it will be the third leading cause of global death by 2030 (7). 

Though Bangladesh has achieved noteworthy progress in its health sector throughout the past decades in comparison with its neighbors(8),it is still suffering from the increased burden of NCDs(9). NCDs account for 61% of the total disease burden in Bangladesh(3). A vast number of people in Bangladesh are suffering from CRDs, where the prevalence of asthma is 5.2% (10) and the prevalence of COPD is 12.5% (7).The World Health Organization (WHO)recommends that NCD care should be integrated into Primary Health Care (PHC) to improve NCD service delivery (2). In response to this recommendation, Bangladesh has set up NCD corners which provide prevention, screening, and treatment for common NCDs at the Upazila sub-district level in its three-tiered PHC system (sub-district, union, and village level)(11)(12).However, there is still a huge gap in accessibility of NCD services in the primary and secondary levels of the health system due to an insufficient government budget for health, which accounts for only 3.4 % of Bangladesh’s GDP (12). In addition to lack of government funding, NCD care in Bangladesh is limited due to poor supply chain management of essential medicines and other commodities (12). Moreover, the health system does not offer most services in rural areas outside of Dhaka, meaning that trained staff and CRD services are inaccessible to patients there(8)

Additionally, there is paucity of data on NCD service readiness, availability, and utilization in primary care facilities due to lack of research in this field as well as weaknesses in the national management information system, DHIS2 (13).Data on NCD service readiness, availability, and utilization in the facilities are crucial for governments and non-governmental organizations (NGOs) to implement appropriate interventions and to assess the quality and impact of the services provided(14). The Service Availability and Readiness Assessment (SARA) is a comprehensive facility-based assessment and WHO SARA Standard Tool is used to collect data for conducting the assessment(15). The tool covers questions on essential services in health facilities, through which service ‘availability’ (whether facilities offer a variety of preventive and curative health services) and ‘readiness’ (whether facilities have the items required to deliver that service at the time of the site visit) can be evaluated rapidly(16). Previously, studies have used the WHO SARA Standard Tool to assess other NCDs in Bangladesh, such as diabetes and cardiovascular disease (CVD)(8). However, WHO SARA Standard Tool has not been utilized to assess CRDs related services within all health facility levels of Bangladesh. 

The Sustainable Development Goal Target 3.4 is to reduce one third premature mortality from non-communicable diseases through prevention and treatment. Universal access to CRD services is an essential prerequisite for achieving this goal.(15).This study will help policy makers address gaps in current CRD service delivery which will further improve action plans for overall NCD prevention and control in Bangladesh.’

--Methods:

1. Generally, this section is not well structured.

Response: Thank you so much for your feedback. It has been fully revised as below-

‘Methods 

Survey Design

This cross-sectional sub-study was part of a larger study called Service Availability and Readiness Assessment (SARA) Survey for NCDs and Disability Service Delivery System in Bangladesh conducted from December 2017 to June 2018. For the purposes of this sub-study one variable for assessing availability and 27 variables for assessing readiness focusing on CRD related services were considered.

WHO SARA Standard Toolcollects information on the availability of medical equipmentrelated to CRD, including their location and functional status and components of support systems (e.g., logistics, maintenance, and management).

Sampling and Recruitment

To identify the total number of hospitals at all three tiers in Bangladesh, the Bangladesh Health Facility Survey (BHFS) 2014, National Institute of Population Research and Training (NIPORT), Management Information System(MIS) Report of Directorate General of Health Services (DGHS), and records from director hospital administration DGHS were used(16). From this, we identified19,184total hospitals in the country.

From this total, the parent study, Service Availability and Readiness Assessment (SARA) Survey for NCDs and Disability Service Delivery System in Bangladesh, selected10 tertiary and specialized hospitals for NCDs, all 60 district hospitals and 61 Mother and Child Welfare Centre, 128 UpazilaHealth Complexes (UHC) (two per district),256 Family Welfare Centres(FWC)& Community Clinics,and 75 NGO and private hospitals. 

This sub-study then excluded 53 health facilities for analysis: specialized hospitals that did not focus on CRD care, such as those devoted solely to cancer; facilities that did not fully complete the questionnaire, and hospitals where no staff member was able to complete the survey.This resulted in a final sample size of 537 facilities depicted in Fig1.

Fig 1: Sampling technique flow chart

The head of each hospital or a management staff member who would have sufficient knowledge of hospital capacity and operations were selected to participate in the survey(Table 2).

Procedures

Forty-eight interviewers (MBBS graduates) were trained for two days to collect data using the survey tooland enter data into REDCap using tablets. The health facilities sampled were contacted beforehand and an appointment was set with the focal person for interviewing. 

Data analysis

Descriptive analyses were performed using the standard core indicators (i.e. staff and guideline, equipment, diagnostic facilities, and medicines) and corresponding tracer indicators (Table 1). 

Calculations for facility “availability” and “readiness” were based upon previous studies that utilized the SARA tool (5) (18). Facility “availability” was assessed by asking if the health facility provided any CRD services (yes or no).

Service readiness was assessed in four stages: 1. Determining the availability of CRD service readiness indicators at each facility level; 2. Calculating the tracer item indexscores (number of tracer item present *100/number of tracer item should be present); 3. Calculating the readiness index (RI) of facilities according to all 4 domains (the mean of all tracer item index scores in each domain); 4. Calculating the facility level’s overall readiness score (the average of the readiness index of all 4 domains). Indices were stratified by facility level and compared to an RI cutoff score of 70%. This cutoff was based upon a study conducted by Wilbroad Mutale et al in Zambia which utilized the SARA tool, where scores above 70% were considered as ‘ready’ to

 manage CRDs at that level (5). All analyses were performed using SPSS Vr 21.0.’

2. Study design most information do not fall under this subsection. For example, training of interviewers, data collection and entry, and etc.

Response: Thank you so much for your feedback. It has been edited as below-

‘This cross-sectional sub-study was part of a larger study called Service Availability and Readiness Assessment (SARA) Survey for NCDs and Disability Service Delivery System in Bangladesh conducted from December 2017 to June 2018. For the purposes of this sub-study one variable for assessing availability and 27 variables for assessing readiness focusing on CRD related services were considered.

WHO SARA Standard Toolcollects information on the availability of medical equipmentrelated to CRD, including their location and functional status and components of support systems (e.g., logistics, maintenance, and management).’

3. Sample size/ sampling technique; rather than providing the references the authors are required to explain in details the sample size calculation as well as sampling procedures.

Response: Thank you so much for your feedback. It has been edited and explained in detail (line 114 to 128). Added figure below (Fig 1)-

Fig 1: Sampling and study inclusion flow chart

4. Data analysis; surprisingly, most of the information written here not telling us how authors analyzed data. The details here are very vague and makes difficult to follow.

Response: Thank you so much for your feedback. It has been revised and explained in details (line 138 to 152)

Also added a reference regarding ‘Cut off value’((line 150, 151) as below-

‘Data analysis

Descriptive analyses were performed using the standard core indicators (i.e. staff and guideline, equipment, diagnostic facilities, and medicines) and corresponding tracer indicators (Table 1). 

Calculations for facility “availability” and “readiness” were based upon previous studies that utilized the SARA tool (5) (18). Facility “availability” was assessed by asking if the health facility provided any CRD services (yes or no).

Service readiness was assessed in four stages: 1. Determining the availability of CRD service readiness indicators at each facility level; 2. Calculating the tracer item indexscores (number of tracer item present *100/number of tracer item should be present); 3. Calculating the readiness index (RI) of facilities according to all 4 domains (the mean of all tracer item index scores in each domain); 4. Calculating the facility level’s overall readiness score (the average of the readiness index of all 4 domains). Indices were stratified by facility level and compared to an RI cutoff score of 70%. This cutoff was based upon a study conducted by Wilbroad Mutale et al in Zambia which utilized the SARA tool, where scores above 70% were considered as ‘ready’ to

 manage CRDs at that level (5). All analyses were performed using SPSS Vr 21.0.’

5. How the cut-off point of 70% reached? Is it based on the authors’ agreement without considering previous literature? If yes, why the authors did not choose to be 80% or 90% or 60%?

Response: Thank you so much for your feedback. It is based upon a previous study on SARA which we have mentioned after revising Methodology section in line 150, 151 like below-

 ‘This cutoff was based upon a study conducted by Wilbroad Mutale et al in Zambia which utilized the SARA tool, where scores above 70% were considered as ‘ready’ to manage CRDs at that level’

1. Ethical statement; This part is missing in the current version.

Response: Thank you so much for your feedback. We have added it in Ethical consideration (line 232 to 235) as below

‘Ethical clearance was obtained from the Ethical Review Board of the Center for Injury Prevention and Research, Bangladesh. Informed written consent was taken from each respondent’ 

--Results:

1. CRD service availability; the text does not tile with the Figure 1 and 2.

Response: Thank you so much for your feedback. Figure 1&2 has been changed into Figure 2&3 respectively. Texts have been edited.

2. CRD service readiness; what is the unit of readiness score? 2nd sentence “….. only 6 tertiary level hospitals out 537 were ready………” the use of 537 facilities in sentence is wrong or the comparison is wrong. The sentence is very confusing.

Response: Thank you so much for your feedback. The unit of readiness score is ‘%’ and it is added.

We deleted the confusing sentences and revised it fully (line 178 to 185) as below-

‘When disaggregated by domains, only tertiary facilities could reach the threshold index in any of the 4 domains(Table 3). Scores in the staff and guideline domain were low, with the lowest by FWC/CC (0.5) (Fig 5).Though tertiary hospitals crossed the cutoff score of 70%, some of them had no pulmonologists (25.0%) nor trained nurses (37.5%) for CRD treatment(Table 3). 

The primary and secondary level of facilities had few diagnostic facilities for CRD with the lowest provided by FWC/CCs(1.3%) (Fig 5). Moreover, 65% of the UHCs did nothave X-ray facilities(Table 3). Except broad spectrum antibiotics, few essential medicines, especially inhalers, were available across facilities. ‘

3. Missing of n in the all figures makes difficult to interpret data.

Response: Thank you so much for your feedback. We have included ‘n’ in all our tables.

.

4. In all figures the test for significance is missing. Therefore, it is difficult to establish the statistical significance of the results.

Response: Thank you so much for your feedback. We did not test significance here. We analyzed data according to the structured WHO SARA tool guideline. 

--Discussion:

1. Generally, the discussion is not well written and the authors failed to discuss in detail the important findings from this study instead they summarized and presented the findings shown in the results section.

Response: Thank you so much for your feedback. We have revised it fully (line 188 to 223). Explained extensively the results, compared it to other studies. Included strength as well as limitation as below-

‘Discussion

The study reveals that, on average, the tertiary hospitals were ready to provide specific service for CRD patients, though there is still significant room for improvement. The rest of the health facilities throughout Bangladesh had inadequate CRD service availability and readiness.

Lack of essential commodities, including medicines and medical supplies, is a constant problem in public health facilities within Bangladesh (12), which was consistent with the limited supply of essential medicines for CRD management found in this study. In particular, supplies of spacers for inhaler and pick flow meter supplies are extremely inadequate.

Existing literature illustrates that Bangladesh has a severe shortage of trained staff and inadequate diagnostic facilities (12), which was principally responsible for facilities’ lower readiness scores in this study. For example, only 50% of the tertiary facilities have a pulmonary technician. The mean readiness index of staff and guideline for DH, UHC, and private hospitals are 16.1%, 15.3%, and 13.2 % respectively, which are similar readiness index scores previously conducted using the SARA tool to assess CVD and diabetes at these facility levels (8). As patient education reduces gaps in health care and disease management, so primary care physicians may be an effective target for interventions to improve CRD management through patient based simulation training in a resource poor country like Bangladesh (17).

While Leslie HH et al.reported, using the SARA Standard Tool, that private facilities in Bangladesh possessed higher RI for overall NCD services in contrast to public facilities (18),this was not true for CRDs in this study (39.5% RI). While according to the interviewee around 90% of DH& UHC (public facilities) and 80% NGO/private facilities provided CRD services to the patients, none of their readiness index scores indicated that they could sufficiently manage patients with CRD. Though Bangladesh’s government established NCD corners in UHCs (11), these facilities’ RI score (33.3%) was less than half of the threshold assessing readiness to treat CRDs (70%).

Almost every indicator was low in the MCWCs and FWC/CCs, which was not wholly surprising. MCWCs and FWCs fall under the Family Planning Wing of the Ministry of Health and Family Welfare, which focuses primarily on maternal and child healthcare, limiting their capacity for CRD care (19). Community Clinics are at the lowest level of healthcare delivery system and do not have adequate supplies nor manpower to provide CRD care to patients, often referring patients to facilities at higher tiers of the PHC system instead (19)(20).

The strength of this study is that we explored the current scenario of CRDs management in facilities throughout Bangladesh stratified by different levels. We explained it both in summarized and disaggregated in all four domains according to WHO SARA Standard Tool. However, there are few limitations such as we could not include all the sample health facilities due to incomplete questionnaire and absence of proper interviewee. Data is based on reported information from the interviewed staff member and was not confirmed by observation. Moreover, we had to use same indicators for all level of facilities as there is no specific evidence mentioning what services should be provided at which level. Also, changes over time could not be observed as it is a cross-sectional study. The available data only reflects on the snapshot of availability and readiness.’

---

## [Decision Letter · Decision Letter 1]

27 Nov 2020

PONE-D-20-06779R1

Assessing Service Availability and Readiness to manage Chronic Respiratory Disease (CRD) in Bangladesh

PLOS ONE

Dear Dr. Paromita,

Thank you for submitting your manuscript to PLOS ONE. After careful consideration, we feel that it has merit but does not fully meet PLOS ONE’s publication criteria as it currently stands. Therefore, we invite you to submit a revised version of the manuscript that addresses the points raised during the review process.

As you will, see the reviewer who re-reviewed the manuscript still has a number of comments on the calculation of the indicators used in the analysis, as well as on the use of English. In addition, I should note that there are still discrepancies in the track changes version and the clean version of the manuscript, which made it difficult to review of manuscript.

We look forward to receiving your revised manuscript.

Kind regards,

David Hotchkiss

Academic Editor

PLOS ONE

Reviewers' comments:

Reviewer's Responses to Questions

**Comments to the Author**

1. If the authors have adequately addressed your comments raised in a previous round of review and you feel that this manuscript is now acceptable for publication, you may indicate that here to bypass the “Comments to the Author” section, enter your conflict of interest statement in the “Confidential to Editor” section, and submit your "Accept" recommendation.

Reviewer #2: (No Response)

2. Is the manuscript technically sound, and do the data support the conclusions?

Reviewer #2: Partly

3. Has the statistical analysis been performed appropriately and rigorously? 

Reviewer #2: No

4. Have the authors made all data underlying the findings in their manuscript fully available?

Reviewer #2: Yes

5. Is the manuscript presented in an intelligible fashion and written in standard English?

Reviewer #2: No

6. Review Comments to the Author

Reviewer #2: There are two versions of the revised paper: a first version commencing on page 19 immediately following the responses to reviewers and a second version headed ‘revised manuscript with track changes’ commencing on page 50.

Unfortunately, while the authors have inserted Table 1 into the revised first version of R1, there is still a discrepancy in lines 221 following in the first version:

The first version of the paper is as follows:

221 The strength of this study is that we explored the current scenario of CRDs management in

222 facilities throughout Bangladesh stratified by different levels. We explained it both in summarized

223 and disaggregated in all four domains according to WHO SARA Standard Tool. However, there

224 are few limitations such as we could not include all the sample health facilities due to incomplete

225 questionnaire and absence of proper interviewee. Also,changes over time could not be observed

226 as it is a cross-sectional study. The available data only reflects on the snapshot of availability and

227 readiness.

The second version of the paper has the following paragraph at the end of the discussion section in the same position as the paragraph above:

228 The strength of this study is that we explored the current scenario of CRDs management in

229 facilities throughout Bangladesh stratified by different levels. We explained it both in

230 summarized and disaggregated in all four domains according to WHO SARA Standard Tool.

231 However, there are few limitations such as we could not include all the sample health facilities

232 due to incomplete questionnaire and absence of proper interviewee. Data is based on reported

233 information from the interviewed staff member and was not confirmed by observation.Moreover,

234 we had to use same indicators for all level of facilities as there is no specific evidence

235 mentioning what services should be provided at which level. Also, changes over time could not

236 be observed as it is a cross-sectional study. The available data only reflects on the snapshot of

237 availability and readiness.

The second version, while not marked in track changes, contains the changes requested in the review, and should replace the same paragraph in the first version.

Responses to questions in the review of the first version of the paper

Q1. Criteria applied to all levels of facilities.

I note the response to my question on different criteria for different levels of facilities: ‘we have used the same indicators for all levels of facilities’. However, even if facilities could not be distinguished on the basis of what services should be offered at what level, calculation of readiness should also take into account whether the facilities claims to offer the service.

However, the availability calculations in this paper appear to be calculated on the basis of all facilities at each level (Table 3 n values) .The WHO SARA guidelines explain that availability should only be calculated on the facilities offering the services. As the proportion of facilities offering the service (reported as availability) was much lower among MCWC and FWC/CCs, the calculation of readiness will be significantly altered.

See reference below:

https://www.who.int/healthinfo/systems/SARA_Implementation_Guide_Chapter8.pdf?ua=1

SARA Implementation guide: page 113

Step 3. Calculate the mean availability of each tracer item The mean availability of each tracer item is equal to the total number of facilities that have the tracer item available (i.e. value=1), divided by the total number of facilities OFFERING THE SERVICE, multiplied by 100 to get a percentage value.

The authors need to address this issue. If the authors choose not to use the SARA guidelines in the calculations, this should be explained; otherwise the calculations need to be repeated based only on facilities that claim that they offer CRD services.

Note that if the readiness scores are calculated based on facilities offering CRD services, then the comments in the discussion lines 212-217 may need to be revised – as the MCWC and FWC facilities also have low proportions who claim to offer CRD services.

Q3 addressed

Q4 There are still issues with syntax and grammar -see errors identified below

Responses to comments

Comments on methods

(1) Additional information on facilities sampled:

The response does not address the issue of additional information on facilities except very briefly. It does not indicate the population served by each facility. Given the very large populations in Bangladesh, it is important that readers understand that a UHC serves a population greater than many districts in other countries. I would recommend a table as initially suggested with average populations per level of health facility.

Methods Table 2

This indicates the % of facilities in the sample, but does not indicate the total number of facilities in the country, so that the sample as a percentage of total facilities can be calculated. This data should be available from the main study. If so a reference to a published paper of the main study would be sufficient to cover this, although inclusion of the data would be preferable.

Line 119-123 states that a selection of facilities was made for the SARA parent study. Please state briefly how facilities were selected.

(2) Comments on results

This point has been adequately addressed

(3) Comments on discussion

See comments above on calculations of service readiness above. This comment has not been adequately addressed.

Comment 2 has been adequately addressed

(4) Comments on conclusions

This comment was addressed in the limitations. However I note that although these changes are noted in the version of the report labelled ‘track changes’ – these changes are not included in the version of the report that appears to be the final revised version.

Table 1, although referred to in the revised version, is not included, although included in the track changes version.

Corrections to syntax

Line 68 health sector (singular)

Line 96 add ‘the’ WHO SARA standard tool

Line 98 – reduce ‘by’ one third

Line 110 – add ‘The’ WHO SARA standard tool

Line 118 – add hospitals ‘and health centres’

Line 124 and following – there is a reference only to hospitals eg ‘the head of each hospital’ – but the survey includes non hospital facilities – suggest replace the word hospital here with facility

Line 134 – ‘and enter data’ – add ‘to’ – and to enter data.

Line 173 – add ‘facility’ – only facility to cross

Line 197 – replace ‘pick’ with ‘peak’

Line 206 – replace ‘true’ with ‘found’

Line 221 - replace ‘few’ with ‘some’ and add ‘to this study’ – there are some limitations to this study

Line 226 – separate Bangladeshis – to ‘Bangladesh is’

7. PLOS authors have the option to publish the peer review history of their article (what does this mean?). If published, this will include your full peer review and any attached files.

Reviewer #2: **Yes: **Krishna Hort

---

## [Author Response · Author response to Decision Letter 1]

1 Feb 2021

Response to Reviewers’ Comments to the Author (2nd version)

Comments to the Author

REVIEWER #2: 

There are two versions of the revised paper: a first version commencing on page 19 immediately following the responses to reviewers and a second version headed ‘revised manuscript with track changes’ commencing on page 50.

Unfortunately, while the authors have inserted Table 1 into the revised first version of R1, there is still a discrepancy in lines 221 following in the first version:

The first version of the paper is as follows:

221 The strength of this study is that we explored the current scenario of CRDs management in

222 facilities throughout Bangladesh stratified by different levels. We explained it both in summarized

223 and disaggregated in all four domains according to WHO SARA Standard Tool. However, there

224 are few limitations such as we could not include all the sample health facilities due to incomplete

225 questionnaire and absence of proper interviewee. Also,changes over time could not be observed

226 as it is a cross-sectional study. The available data only reflects on the snapshot of availability and

227 readiness.

The second version of the paper has the following paragraph at the end of the discussion section in the same position as the paragraph above:

228 The strength of this study is that we explored the current scenario of CRDs management in

229 facilities throughout Bangladesh stratified by different levels. We explained it both in

230 summarized and disaggregated in all four domains according to WHO SARA Standard Tool.

231 However, there are few limitations such as we could not include all the sample health facilities

232 due to incomplete questionnaire and absence of proper interviewee. Data is based on reported

233 information from the interviewed staff member and was not confirmed by observation. Moreover,

234 we had to use same indicators for all level of facilities as there is no specific evidence

235 mentioning what services should be provided at which level. Also, changes over time could not

236 be observed as it is a cross-sectional study. The available data only reflects on the snapshot of

237 availability and readiness.

The second version, while not marked in track changes, contains the changes requested in the review, and should replace the same paragraph in the first version.

Response: Thank you so much for your feedback. We have addressed your concern and replaced the paragraph containing the changes in line (228-235), at the end of the discussion in the revised manuscript without track change mode as below- 

“The strength of this study is that we explored the current scenario of CRDs management in facilities throughout Bangladesh stratified by different levels. We explained it both in summarized and disaggregated in all four domains according to WHO SARA Standard Tool. However, there are few limitations such as we could not include all the sample health facilities due to incomplete questionnaire and absence of proper interviewee. Data is based on reported information from the interviewed staff member and was not confirmed by observation. Moreover, we had to use same indicators for all level of facilities as there is no specific evidence mentioning what services should be provided at which level.”

Responses to questions in the review of the first version of the paper

Q1. Criteria applied to all levels of facilities.

I note the response to my question on different criteria for different levels of facilities: ‘we have used the same indicators for all levels of facilities’. However, even if facilities could not be distinguished on the basis of what services should be offered at what level, calculation of readiness should also take into account whether the facilities claims to offer the service.

However, the availability calculations in this paper appear to be calculated on the basis of all facilities at each level (Table 3 n values) .The WHO SARA guidelines explain that availability should only be calculated on the facilities offering the services. As the proportion of facilities offering the service (reported as availability) was much lower among MCWC and FWC/CCs, the calculation of readiness will be significantly altered.

See reference below:

https://www.who.int/healthinfo/systems/SARA_Implementation_Guide_Chapter8.pdf?ua=1

SARA Implementation guide: page 113

Step 3. Calculate the mean availability of each tracer item The mean availability of each tracer item is equal to the total number of facilities that have the tracer item available (i.e. value=1), divided by the total number of facilities OFFERING THE SERVICE, multiplied by 100 to get a percentage value.

The authors need to address this issue. If the authors choose not to use the SARA guidelines in the calculations, this should be explained; otherwise the calculations need to be repeated based only on facilities that claim that they offer CRD services.

Note that if the readiness scores are calculated based on facilities offering CRD services, then the comments in the discussion lines 212-217 may need to be revised – as the MCWC and FWC facilities also have low proportions who claim to offer CRD services.

Response to Q1: Thank you so much for your feedback. We have addressed the issue and re-analyzed our data. We rigorously did literature review and found that actually CRD services are provided upto Upazila (UHC) level only (Biswas et al) and MCWC, FWC provide only family planning services. So, we excluded MCWC (61) & USC,FWC, CC (214) from our analysis which made our sample size into 262. We also edited these in methodology and inserted the following paragraph (line 114-137) in the revised manuscript without track change mode -

“In Bangladesh, public facilities are established according to its administrative units (division, district, upazila, union) where tertiary, secondary, and primary facilities are situated in the divisions (medical college hospitals), districts (DH, MCWC), upazila (UHC), union (FWC, CC) respectively [17]. As the number of population living in administrative units is different, so the population served by each facility is not defined. It may be mentioned that UHCs provide 31% of government health service and each of it serves 100,000 to 400,000 people depending on its size [18]. 

To identify the total number of hospitals at all three tiers in Bangladesh, the Bangladesh Health Facility Survey (BHFS) 2014, National Institute of Population Research and Training (NIPORT), Management Information System(MIS) Report of Directorate General of Health Services (DGHS), and records from director hospital administration DGHS were used (16). From this, we identified19,184 total hospitals and health centers in the country [19] .

From this total, the parent study, Service Availability and Readiness Assessment (SARA) 

Survey for NCDs and Disability Service Delivery System in Bangladesh, selected 590 facilities using a stratified random sampling procedure according to administrative units and level of facilities (Fig1). CRD services are provided from tertiary level of health facilities upto upazila level health facilities in Bangladesh. [8] This includes UHCs, district hospitals, and medical college hospitals. Conversely, MCWCs and FWCs were excluded from this sub study since these facilities only offer family planning services, and not CRD services. Therefore, this sub study found a total of 273 facilities that provided CRD services. However, 11 more facilities were excluded from this sub study during the analysis of data. These facilities were excluded either because the questionnaires were incomplete or the hospitals were specialized in diseases other than CRD and were therefore irrelevant for this current study. The resulting sample size of 262 facilities is depicted in Fig1.”

Fig 1: Sampling and study inclusion flow chart

[Figure Inserted in the responses to Reviewer's documents]

N>B- UHC: Upazila Health Complex, MCWC: Maternal and Child Welfare Center, FWC: Family Welfare Center, CC: Community Clinic, NGO: Non Governmaent Organization.

*rest of the district hospitals had been converted to medical college hospital (tertiary facilities)”

Also, we followed SARA guideline for calculating mean availability (Table 2). For example among 8 tertiary facilities, Pulmonologists are available in 6. So, the mean availability is 75.0%.

Then we calculated the readiness scores based on facilities offering CRD services as we have already excluded MCWC, FWC, CC (Fig 2&Fig 3).

We also changed it in methodology in line 156-165 in the revised manuscript without track change mode like below-

“Mean availability and Service readiness were assessed in three stages: 1. Determining the mean availability of CRD service readiness indicators/tracer item at each facility level(number of facilities that have the tracer item available *100/ the total number of facilities); 2. Calculating the readiness index (RI) of facilities according to all 4 domains (the mean of all tracer item availability score in each domain);3. Calculating the facility level’s overall readiness score (the average of the readiness index of all 4 domains). Indices were stratified by facility level and compared to an RI cutoff score of 70%. This cutoff was based upon a study conducted by Wilbroad Mutale et al in Zambia which utilized the SARA tool, where scores above 70% were considered as ‘ready’ tomanage CRDs at that level (5). All analyses were performed using SPSS Vr 21.0.”

We have also deleted the paragraph describing MCWC,FWC in discussion.

Q3 addressed

Q4 There are still issues with syntax and grammar -see errors identified below

Response: Thank you for your feedback. We have addressed these issues mentioned below.

Line 68 health sector (singular)- corrected in line 67 in the revised manuscript without track change mode

Line 96 add ‘the’ WHO SARA standard tool- corrected in line 96 in the revised manuscript without track change mode

Line 98 – reduce ‘by’ one third- corrected in line 98 in the revised manuscript without track change mode

Line 110 – add ‘The’ WHO SARA standard tool—corrected in line 110 in the revised manuscript without track change mode

Line 118 – add hospitals ‘and health centres’—corrected in line 125 in the revised manuscript without track change mode

Line 124 and following – there is a reference only to hospitals eg ‘the head of each hospital’ – but the survey includes non hospital facilities – suggest replace the word hospital here with facility -corrected in line 142 in the revised manuscript without track change mode

Line 134 – ‘and enter data’ – add ‘to’ – and to enter data.—corrected in line 146 in the revised manuscript without track change mode

Line 173 – add ‘facility’ – only facility to cross---- corrected in line 192 in the revised manuscript without track change mode

Line 197 – replace ‘pick’ with ‘peak’ --- corrected all over the manuscript, Table 1, Table 2 in the revised manuscript without track change mode

Line 206 – replace ‘true’ with ‘found’—corrected in line 222 in the revised manuscript without track change mode

Line 221 - replace ‘few’ with ‘some’ and add ‘to this study’ – there are some limitations to this study---- corrected in line 231 in the revised manuscript without track change mode

Line 226 – separate Bangladeshis – to ‘Bangladesh is’--- corrected in line 238 in the revised manuscript without track change mode

Responses to comments

Comments on methods

(1) Additional information on facilities sampled:

The response does not address the issue of additional information on facilities except very briefly. It does not indicate the population served by each facility. Given the very large populations in Bangladesh, it is important that readers understand that a UHC serves a population greater than many districts in other countries. I would recommend a table as initially suggested with average populations per level of health facility.

Response: Thank you for your feedback. We have inserted the following paragraph in methods line (114-120) in the revised manuscript without track change mode

“In Bangladesh, public facilities are established according to its administrative units (division, district, upazila, union) where tertiary, secondary, and primary facilities are situated in the divisions (medical college hospitals), districts (DH, MCWC), upazila (UHC), union (FWC, CC) respectively [17]. As the number of population living in administrative units is different, so the population served by each facility is not defined. It may be mentioned that UHCs provide 31% of government health service and each of it serves 100,000 to 400,000 people depending on its size [18].”

Methods Table 2

This indicates the % of facilities in the sample, but does not indicate the total number of facilities in the country, so that the sample as a percentage of total facilities can be calculated. This data should be available from the main study. If so a reference to a published paper of the main study would be sufficient to cover this, although inclusion of the data would be preferable.

Line 119-123 states that a selection of facilities was made for the SARA parent study. Please state briefly how facilities were selected.

Response: Thank you for your feedback. We have included the required changes in methods like below (line 121-137) in the revised manuscript without track change mode

“To identify the total number of facilities at all three tiers in Bangladesh, the Bangladesh Health Facility Survey (BHFS) 2014, National Institute of Population Research and Training (NIPORT), Management Information System(MIS) Report of Directorate General of Health Services (DGHS), and records from director hospital administration DGHS were used (16). From this, we identified 19,184 total hospitals and health centers in the country [19] .

From this total, the parent study, Service Availability and Readiness Assessment (SARA) 

Survey for NCDs and Disability Service Delivery System in Bangladesh, selected 590 facilities using a stratified random sampling procedure according to administrative units and level of facilities (Fig1). CRD services are provided from tertiary level of health facilities upto upazila level health facilities in Bangladesh. [8] This includes UHCs, district hospitals, and medical college hospitals. Conversely, MCWCs and FWCs were excluded from this sub study since these facilities only offer family planning services, and not CRD services. Therefore, this sub study found a total of 273 facilities that provided CRD services. However, 11 more facilities were excluded from this sub study during the analysis of data. These facilities were excluded either because the questionnaires were incomplete or the hospitals were specialized in diseases other than CRD and were therefore irrelevant for this current study. The resulting sample size of 262 facilities is depicted in Fig1.. 

Fig 1: Sampling and study inclusion flow chart

 [Figure Inserted in the responses to Reviewer's documents]

N>B- UHC: Upazila Health Complex, MCWC: Maternal and Child Welfare Center, FWC: Family Welfare Center, CC: Community Clinic, NGO: Non Governmaent Organization.

*rest of the district hospitals had been converted to medical college hospital (tertiary facilities)”

As the parent study is still not published, so we put the reference of the Bangladesh Health Facility Survey (BHFS) 2014 from where we collected the facility list.

(2) Comments on results

This point has been adequately addressed

(3) Comments on discussion

See comments above on calculations of service readiness above. This comment has not been adequately addressed.

Response: Thank you so much for your feedback. We have changed the paragraph in discussion (excluded the portion of MCWC,FWC,CC). Also followed SARA tool calculating mean availability (table 2) and readiness (Fig 2,3)

Comment 2 has been adequately addressed

(4) Comments on conclusions

This comment was addressed in the limitations. However I note that although these changes are noted in the version of the report labelled ‘track changes’ – these changes are not included in the version of the report that appears to be the final revised version.

Table 1, although referred to in the revised version, is not included, although included in the track changes version.

Response: Thanks for addressing this issue. We have included these both in track change and clean version of the manuscript.

Corrections to syntax

Response:

Line 68 health sector (singular)- corrected in line 67 in the revised manuscript without track change mode

Line 96 add ‘the’ WHO SARA standard tool- corrected in line 96 in the revised manuscript without track change mode

Line 98 – reduce ‘by’ one third- corrected in line 98 in the revised manuscript without track change mode

Line 110 – add ‘The’ WHO SARA standard tool—corrected in line 110 in the revised manuscript without track change mode

Line 118 – add hospitals ‘and health centres’—corrected in line 125 in the revised manuscript without track change mode

Line 124 and following – there is a reference only to hospitals eg ‘the head of each hospital’ – but the survey includes non hospital facilities – suggest replace the word hospital here with facility -corrected in line 142 in the revised manuscript without track change mode

Line 134 – ‘and enter data’ – add ‘to’ – and to enter data.—corrected in line 146 in the revised manuscript without track change mode

Line 173 – add ‘facility’ – only facility to cross---- corrected in line 192 in the revised manuscript without track change mode

Line 197 – replace ‘pick’ with ‘peak’ --- corrected all over the manuscript, Table 1, Table 2 in the revised manuscript without track change mode

Line 206 – replace ‘true’ with ‘found’—corrected in line 222 in the revised manuscript without track change mode

Line 221 - replace ‘few’ with ‘some’ and add ‘to this study’ – there are some limitations to this study---- corrected in line 231 in the revised manuscript without track change mode

Line 226 – separate Bangladeshis – to ‘Bangladesh is’--- corrected in line 238 in the revised manuscript without track change mode.

---

## [Decision Letter · Decision Letter 2]

12 Feb 2021

Assessing Service Availability and Readiness to manage Chronic Respiratory Disease (CRD) in Bangladesh

PONE-D-20-06779R2

Dear Dr. Paromita,

We’re pleased to inform you that your manuscript has been judged scientifically suitable for publication and will be formally accepted for publication once it meets all outstanding technical requirements.

Kind regards,

David Hotchkiss

Academic Editor

PLOS ONE

Additional Editor Comments (optional):

Reviewers' comments:

Reviewer's Responses to Questions

**Comments to the Author**

1. If the authors have adequately addressed your comments raised in a previous round of review and you feel that this manuscript is now acceptable for publication, you may indicate that here to bypass the “Comments to the Author” section, enter your conflict of interest statement in the “Confidential to Editor” section, and submit your "Accept" recommendation.

Reviewer #2: All comments have been addressed

2. Is the manuscript technically sound, and do the data support the conclusions?

Reviewer #2: Yes

3. Has the statistical analysis been performed appropriately and rigorously? 

Reviewer #2: Yes

4. Have the authors made all data underlying the findings in their manuscript fully available?

Reviewer #2: Yes

5. Is the manuscript presented in an intelligible fashion and written in standard English?

Reviewer #2: Yes

6. Review Comments to the Author

Reviewer #2: (No Response)

7. PLOS authors have the option to publish the peer review history of their article (what does this mean?). If published, this will include your full peer review and any attached files.

Reviewer #2: **Yes: **Krishna Hort

---

## [Editor Report · Acceptance letter]

22 Feb 2021

PONE-D-20-06779R2 

Assessing service availability and readiness to manage Chronic Respiratory Diseases (CRDs) in Bangladesh 

Dear Dr. Paromita:

I'm pleased to inform you that your manuscript has been deemed suitable for publication in PLOS ONE. Congratulations! Your manuscript is now with our production department. 

Kind regards, 

on behalf of

Dr. David Hotchkiss 

Academic Editor

PLOS ONE